# A multi-phase project to develop a patient-reported measure of barriers to antiretroviral therapy adherence for use in HIV care: The 7-Item I-Score

Kim Engler[1]*, David Lessard[1], Serge Vicente[2], Darrell H. S. Tan[3,4,5], Karine Lacombe[6,7], Bertrand Lebouché[1,8]

**1** Center for Outcomes Research and Evaluation, Research Institute of the McGill University Health Centre, Montreal, Quebec, Canada, **2** Department of Mathematics and Statistics, University of Montreal, Montreal, Quebec, Canada, **3** Division of Infectious Diseases, St. Michael's Hospital, Toronto, Canada, **4** Department of Medicine, University of Toronto, Toronto, Canada, **5** Institute of Health Policy, Management and Evaluation, University of Toronto, Toronto, Canada, **6** Sorbonne Université, Inserm, Institut Pierre Louis d'Épidémiologie et de Santé Publique (iPLESP), Paris, France, **7** Assistance Publique-Hôpitaux de Paris (AP-HP), Groupe Hospitalo-universitaire Sorbonne Université, Hôpital Saint-Antoine, Service de Maladies Infectieuses et Tropicales, Paris, France, **8** Department of Family Medicine, McGill University, Montreal, Quebec, Canada

☯ These authors contributed equally to this work.
* kimcengler@gmail.com

## Abstract

Adherence to antiretroviral therapy (ART) is a priority in HIV care. Our goal was to develop and validate a new, short patient-reported outcome measure to screen for barriers to daily oral ART adherence in HIV care in Canada and France. The overarching design was that of a multi-phase, multi-site instrument development project. It involved a previously published qualitative research synthesis (to develop a conceptual framework from which to generate items). Here, we present the results of cognitive interviews (for content validity testing of the preliminary items) and of a longitudinal survey-based study where the revised 7-item instrument (the I-Score) was administered at two timepoints (baseline, 4 weeks), along with five indirect measures of ART adherence, including self-reported HIV viral load and, for a subset of Montreal participants, plasma viral load. The survey data was used to analyze the instrument's measurement properties, namely structural validity (e.g., with Pearson inter-item correlations); construct validity, including cross-cultural validity, for a variety of sociodemographic groups (with receiver operating curve (ROC) analyses and areas under the curve (AUC); reliability (with the intraclass correlation coefficient (ICC)); and measurement error (with the standard error of measurement (SEM)). Study participants were adults living with HIV on antiretroviral therapy recruited from hospital-based infectious disease centers and community-based organizations in Montreal, Toronto, and Paris. The qualitative synthesis of 41 studies led to a framework composed of 6 barrier domains. One item was generated per domain. The

**Data availability statement:** Data is available on Open Science Framework (https://osf.io/942ug/).

**Funding:** This work received support from the Canadian Institutes of Health Research (CIHR) HIV Clinical Trials Network (CTN) (CTN283). It was conducted within the program of a Strategy for Patient-Oriented Research (SPOR) Mentorship Chair in Innovative Clinical Trials awarded by the CIHR to BL (Grant number 383427). It benefited from methodological expertise and funding from the Quebec SPOR Support Unit -McGill Methodological Developments Platform (Grant number M006). It was also supported, in part, by a research grant from the Investigator Initiated Studies Program of Merck Canada Inc. (Grant number IISP-53538), which had no hand in the design, conduct, or writing up of this work. French (France) participation was supported, in part, by funding from MSD Avenir awarded to KL and BL and managed by the AP-HP Foundation (Grant number MSDAVENIR DS-2018-0072). The opinions expressed in this study are those of the authors and do not necessarily represent those of Merck Canada Inc or its affiliates or related companies. BL is supported by two career awards: a Fonds de Recherche du Québec-Santé (FRQ-S) Senior Research Scholar award (#311200) and the LE 250, from Quebec's Ministry of Health for researchers in family medicine. From 2021 to 2023, SV was supported by a CTN Postdoctoral Fellowship Award. DHST is supported by a Tier 2 Canada Research Chair in HIV Prevention and STI Research. The funders had no role in study design, data collection and analysis, decision to publish, or preparation of the manuscript.

**Competing interests:** BL has received research support and consulting fees from ViiV Healthcare, Merck, and Gilead. KL has received travel support and consulting fees from ViiV Healthcare, Merck, and Gilead. DHST's institution has received research support for investigator-initiated research studies from Abbvie and Gilead, and for participation in industry-sponsored trials from Glaxo Smith Kline. The other authors have no competing interests to declare. This does not alter our adherence to PLOS ONE policies on sharing data and materials.

6-item measure was submitted to cognitive testing with 12 adults living with HIV, leading to several changes, including the division of one item into two, creating a 7-item measure. Overall, 305 adults living with HIV participated in the survey. Inter-item correlations were low to moderate, suggesting no redundant items. Among conclusive results, AUC's were all above the predefined threshold (≥ 0.70) for self-reported viral load and plasma viral load for the global sample and across all subgroups examined. The ICC, at 0.81, was also above the predefined threshold (≥ 0.70). The SEM was 0.43. Overall, the evidence generated supports the 7-item I-Score measure's content validity, construct validity/cross-cultural validity, reliability, and acceptable measurement error, in its specified context of use.

## Introduction

The prevention and management of HIV infection continues to be a pressing concern worldwide. In Canada, the Public Health Agency estimated the number of people living with HIV in 2020 to be over 62,000 [1] and recorded a 25% increase in new diagnoses, from 2021 to 2022 [2]. In Montreal (Quebec), the second most populous city in the country, new HIV diagnoses more than doubled over the same period [3].

Antiretroviral therapy (ART) is crucial to control HIV infection and its onward transmission. However, people living with HIV report a wide range of impediments to taking their medication [4] and almost half of adults with HIV in Canada (45%) are estimated to have suboptimal ART adherence [5], comparable to global ART adherence estimates [6, 7]. Healthcare providers are not always accurate judges of patient adherence [8] and HIV viral load is often used as a proxy in clinical practice [9]. Importantly, most people living with HIV on ART in Canada (95%) are estimated to have achieved the treatment goal of viral suppression (i.e., with a last HIV RNA test result of <200 copies/ml) [1], again, similar to global estimates [10]. Nevertheless, concerns remain not only for the residual 5%, but for sub-population differences in the attainment of viral suppression, with lower levels observed among females, individuals who inject drugs, and Indigenous peoples in Canada [1]. Additional concerns include attrition over time among the virally suppressed [11]; unplanned treatment interruptions [12] or cessation [13]; and the role of adherence in persistent low-level viremia (e.g., consecutive viral loads ≥50 copies/ml) and the latter's clinical implications, such as ART resistance [14, 15].

Adherence is a recognized priority for HIV service delivery [16]. However, there is no consensus and limited guidance on the most reliable way to identify adherence challenges, including among patients engaged in care models involving less interaction with the healthcare system [16]. Overall, these observations highlight the complexity of adherence, potential gaps in clinical care, and the need for innovative solutions. Furthermore, it seems especially urgent to emphasize a preventative approach, screening for adherence barriers before they impact clinical outcomes such as viral load. Patient-reported outcome measures could fulfill this function and potentially also contribute to more person-centered ART adherence management.

## The I-Score study

As a part of the Canadian HIV Clinical Trials Network (CTN) study CTN283, our team developed a short, 7-item patient-reported instrument (The I-Score) to measure perceived ART adherence barriers. It was designed for use in routine, adult HIV care to foster systematic and streamlined assessment of these barriers and improve their management. The "I" in I-Score refers to the concept of "minimal interference" in life to summarize HIV expert opinions on characteristics of the ideal antiretroviral therapy, as analyzed by our team in earlier work [17], which helped inspire the measure's development. The I-Score was mainly intended for electronic administration in clinical practice (for example, via a patient portal) in Montreal (Quebec) and Paris (France) among adults with HIV on daily oral ART. However, hard copy administration remains an option.

Further justification for the development of this measure comes from our team's evaluation of existing patient-reported measures of barriers to ART adherence. This work found that they lacked comprehensiveness and mainly assessed reasons for missed doses [18], limiting their utility as clinical tools to prevent the impacts of non-adherence. Creating our measure was also consistent with the results of our needs assessment for the features of a patient portal among people living with HIV and healthcare professionals (n = 114) in the target settings [19]; this survey-based assessment identified that 90% or more of both groups were interested in portal-based administration of patient-reported measures focused on the experience of treatment and HIV self-management, categories within which the I-Score falls.

## Methods

Over the years, this multi-phase, multi-site instrument development project (the I-Score Study) has generated evidence to support the tool's validity for our context of use. With this manuscript, our overarching objectives are to describe this patient-reported outcome measure's (PROM) development process, emphasizing the original research underlying its most recent results. Specifically, we will summarize the creation of our previously published conceptual framework and present the methods and new findings related to the measure's cognitive testing and measurement properties. The instrument was produced in English and subsequently translated to French, and both language versions were evaluated for suitability for use in both Canadian/Quebec French or English and France French contexts. Details on our translation approach are published elsewhere [20]. The general components of the I-Score's development process are presented in Fig 1.

**Ethics statements. Patient consent for publication:** Not applicable.

**Ethics approval:** The I-Score Study obtained ethics approval from the Research Ethics Board (REB) of the McGill University Health Centre (MUHC) in Montreal, Quebec, Canada (original approval received December 1, 2014: 2015–2331, 14–229 PSY). In addition to annual renewals until November 29, 2023, two subsequent amendments were also approved by the MUHC REB (an amendment to the I-Score measure's validation plan: approved January 18, 2022; an amendment for the retrospective analysis of plasma viral load test results of patients with HIV at the Chronic Viral Illness Service of the McGill University Health Centre, waiving the requirement for informed consent: approved September 21, 2023). For the Toronto site (St. Michael's Hospital), REB approval was originally obtained from Unity Health Toronto, Toronto, Ontario, Canada, on November 10, 2016 (REB# 16–104) and renewed until November 10, 2023. For the affiliated French sites, approval was obtained from the REB of the Nantes University Health Centre in Nantes, France, on March 6, 2018 (RC17_0310 N° EUDRACT: 2017-A02460-53; French legislation requires no subsequent annual renewals). The project to meaningfully involve people living with HIV in the I-Score Study, in part through a patient advisory committee, was approved by the MUHC REB on September 8, 2015 (project number: 15–188-MUHC).

**Patient and public involvement.** From this project's inception, people living with HIV have been regularly consulted as advisory committee members [21] and/or as patient-partners (e.g., co-author SV). They have been repeatedly and meaningfully engaged in the planning (e.g., giving input on data collection tools), conduct (e.g., participating in and providing feedback on analyses), and knowledge translation of the work produced through this project. Involved people

**Justification for the new measure**

- Review of existing patient-reported measures of HIV medication adherence barriers (n=31 instruments)
- Needs assessment with people with HIV and healthcare professionals (n=114) of patient-reported measures to administer through a patient portal
- I-Score Consulting Team (patient advisory committee) feedback (n=10 members)

**Conceptual framework/item development**

- Qualitative synthesis (n=41 studies) of barriers to ART adherence among adults with HIV in developed countries, leading to a 6-domain framework
- Items generated to capture each domain of the framework

**Cognitive testing***

- Cognitive testing of the draft I-Score measure with people with HIV (n=12), assessing comprehensibility, comprehensiveness, relevance, and acceptability as to the:
  - Instructions
  - Response options
  - Recall period
  - Items
  - Length
- To assess content validity

**Assessment of measurement properties***

- Online survey at baseline and 4 weeks later administering the I-Score and additional measures to people with HIV (n=305)
- Extraction of viral load test results of participants at the Chronic Viral Illness Service (Montreal, Quebec)
- To assess reliability, measurement error, and construct validity, in part, through:
  - Hypothesis testing including for cross-cultural validity

**Fig 1. The 7-item I-Score's development process to date and its associated objectives and methods.** The asterisks indicate steps which have not previously been published and are the focus of this manuscript. Figure template by PresentationGO.com.

living with HIV have co-authored related peer-reviewed publications on their experience [22, 23] and presented at HIV conferences. All participating people with HIV who were part of our involvement activities in the I-Score Study gave their informed and written consent.

## The conceptual framework

To inform item generation (content elicitation) for the I-Score, a previously published conceptual framework was developed by the team [24] which we will briefly describe. For this purpose, as we were especially interested in the patient perspective on barriers to ART adherence, we conducted, with a librarian-guided search strategy, a qualitative synthesis (using thematic analysis) of 41 studies employing qualitative methods among adults living with HIV on ART in developed countries [24]. The resultant conceptual framework has six domains and 20 subdomains of barriers (For a color figure of the framework, see [25]). The six domains are: Cognitive and emotional aspects ("Your thoughts and feelings"), Lifestyle factors ("Your activities"), Social and material context ("Your situation"), Characteristics of ART ("Your medication"), Health experience and state ("Your health"), and Healthcare services and system ("Your care"). The calculated saturation levels show that from 66% to 100% of included qualitative studies among people with HIV reported barriers within each domain

[24]. Interestingly, our framework domains overlap considerably with those of the 2003 World Health Organization (WHO) model of factors affecting adherence to long-term therapy [26]. However, for our needs, we wished to produce a framework that builds specifically on patient views of barriers (i.e., from qualitative research with people living with HIV), incorporates more current research, has a far greater level of granularity, and is specific to our context (e.g., adults living with HIV in developed countries).

When we mapped the content of existing patient-reported measures of barriers to ART adherence to our conceptual framework, we observed that, on average, they covered 4 of our 6 domains and only 7 of our 20 subdomains [18]. The least frequently addressed was the Healthcare services and system domain, with only a fifth (22%) of the 31 instruments reviewed including any related content. This is consistent with the limited comprehensiveness observed in generic measures of medication adherence barriers [27] and the relative underrepresentation of healthcare team and system-related barriers, across adherence barrier measures [28]. Average length of the reviewed instruments was also 14 items which may be longer than ideal for use across care settings [9].

While our qualitative synthesis identified over 70 distinct barriers to ART adherence [24], a lengthy measure would not be feasible for routine administration in HIV care (while it may be more appropriate for research purposes which our team intends to explore). For this specific context of use, we therefore drafted one item per framework domain to essentially capture the presence of barriers within a given domain. We defined barrier domains as "areas of life" that "make it difficult to take" ART and difficulty was defined as medication-taking that "was either skipped, delayed or very unpleasant." We led several rounds of feedback related to this measure (with representation of healthcare professionals, measurement specialists, people living with HIV, and research staff) including meetings with our patient advisory committee (on the conceptual framework: September 27, 2017; on the measure: March 25, 2020) and among team members on the measure (e.g., March 17, 2020; June 19, 2020). Then, the English and French measures were submitted to cognitive testing.

## Cognitive testing

### Objectives and methods

The main goals of cognitive testing were to assess the measure's comprehensiveness, relevance, and comprehensibility from the perspective of people living with HIV, to ensure content validity. Purposive sampling within adult clinical settings (Chronic Viral Illness Service-McGill University Health Centre in Montreal, Quebec and Service des maladies infectieuses et tropicales-Saint-Antoine Hospital, Assistance Publique-Hôpitaux de Paris (AP-HP), in Paris, France) was used to identify 12 diverse cognitive testing participants to ensure adequate representation of the three main interest groups (4 participants each): 1) francophone Quebeckers; 2) anglophone Quebeckers; and 3) francophone Parisians. These recruitment targets were based on the recommended sample size requirements for qualitative content analysis studies of patient-reported measures, as indicated by the COnsensus-based Standards for the selection of health Measurement INstruments (COSMIN) group [29]. They specify that a sample of at least 7 participants is deemed 'very good' (the best rating; achieved for our global sample), and 4–6 participants is considered 'adequate' (achieved for each main subgroup). A minimum of 3 female participants was set to sufficiently represent this gender group which accounts for 25% of people living with HIV in Canada [30].

A qualified member of the research team obtained written informed consent and conducted the semi-structured interviews (one-on-one or by focus group) in either English or French (DL), following a predefined script (S1 File). These were performed to receive input on the measure's content, including its instructions, the six items, response format, and recall period; to assess comprehension; and to gauge perceived comprehensiveness, relevance, and acceptability. All individual interviews and focus groups were audio-recorded and transcribed verbatim. Content analysis was used to categorize the information collected and identify areas for improvement. Independent coding by two individuals was done with one focus group transcript to calibrate the coding scheme. Analysis was performed with Taguette version 1.4.1 [31], a free, open-source qualitative research tool.

## Results

Cognitive testing was conducted from September 23 to November 16, 2020. This involved three focus groups with from 2 to 4 participants and three individual interviews. Each lasted from 102 to 133 minutes in duration.

Participants included nine males and three females, all born between 1992 and 1956. Some descriptive data was missing for one male. Eight lived in the Montreal area (Quebec) and four in Paris (France). Six participants identified as homosexual, four as heterosexual and one as bisexual. Four were anglophone and the remainder, francophone. More than half (n = 7) reported university as their highest level of education, while three reported college and one, high school. Half reported immigrating to their country of residence. None had ever injected drugs, and all were satisfied with their current HIV medication. The participants had been living with HIV from a little over 1–35 years (M = 13.3, SD = 11.8).

Areas for improvement were identified for item ordering, item content, the response scale, the instructions, and the recall period. These are summarized in Table 1. The analyses were discussed at a team meeting, leading to the implemented changes described. Notably, after this stage, the measure increased by 1 item to 7 items.

## Measurement properties

The 7-item I-Score corresponds to a formative measurement model, where, for instance, the items are not expected to be highly correlated or interchangeable. Therefore, some sources of evidence (e.g., related to internal consistency) are not required to establish measure quality [32].

## Objectives

Our specific objectives, at this point, were to: assess the 7-item I-Score's structural validity, concurrent and predictive validities, reliability, and measurement error. We also considered measurement invariance and cross-cultural validity by examining the influence of several sociodemographic factors and mode of survey administration on evidence of the I-Score's concurrent and predictive validities.

## Methods

**Design and sample.** This online survey-based study followed a prospective, longitudinal design with two periods of observation (Time 1: baseline and Time 2: Week 4). Eligible participants self-identified as at least 18 years old, living with HIV, currently on antiretroviral therapy, and a resident of Canada or France. At least 25% of the sample needed to be composed of self-identified women (whether cis- or trans- gender) to ensure their adequate representation [30, 33]. Convenience sampling was used. Participants were recruited by staff at participating healthcare settings (McGill University Health Centre (Montreal, Canada); St. Michael's Hospital (Toronto, Canada); Saint-Antoine Hospital (Paris, France)) and HIV community-based organizations (e.g., Gap-Vies, Portail VIH/Sida du Québec (Montreal, Canada)). Written informed consent was obtained from all participants. The targeted sample size was based on a 5:1 ratio of participants to evaluated measure items, deemed adequate by some statisticians for principal components analysis (PCA) [34, 35] (5 participants x 49 items = 245 participants). Note that along with the 7-item I-Score which assesses barrier domains, 42 additional items on specific barriers identified by our qualitative synthesis [24] and retained from a Delphi consensus process [20, 25] were administered as well and will be eventually submitted to PCA to develop additional instruments for research and/or care.

**Data collection.** The surveys were built and managed on Research Electronic Data Capture (REDCap) [36], a secure web application. From January 19, 2022, to February 21, 2023, participants completed the 7-item I-Score, at baseline (Time 1) and 4 weeks later (Time 2). Note that recruitment from the Toronto site began November 14, 2022. Along with the 7-item I-Score, four additional self-reported adherence-related variables were measured, for hypothesis testing, at each time point: 1) ART adherence in the last 30 days, using a continuous scale from 0% to 100% [37]; 2) ART adherence in the last 7 days [38], using an ordinal scale from 1 (=all HIV pills taken) to 5 (=no HIV pills taken); 3) intention to adhere to

Table 1. Changes implemented to the I-Score following cognitive testing.

| Aspect | Summary of participant reasoning (when provided) | Cognitive interview session* | | | | | | Implemented change |
|---|---|---|---|---|---|---|---|---|
| | | 1 | 2 | 3 | 4 | 5 | 6 | |
| **Response format** | Gradations in the response scale would be better (from 0 or 1–10) as they are customary, facilitate answering and/or allow for greater precision. | X | | X | | | | We transformed the response format from a continuum to an 11-point scale from 0 to 10. |
| **Instructions** | Gradations in the response scale would be better (from 0 or 1–10) as they are customary, facilitate answering and/or allow for greater precision. | X | | X | | | | We revised the instructions to match the new response format (from "Mark your response on the line" to "Answer with a number from 0 (Never made it difficult) to 10 (Always made it difficult)." |
| **Recall period** | The timeframe (i.e., the past 4 weeks) is too short to capture some barriers (e.g., winter travel) or for experienced treatment-takers. Other timeframes are preferable (e.g., past year, past 6 months, past 3 months, past 2 months, since your last consultation). | X | X | | | X | | We will reconsider the measure's recall period when next implementing the I-Score in HIV care. |
| **Item order** | All other answers depend on the acceptance or not of one's HIV status [which is addressed in item 2, "Your thoughts and feelings"]. | | | X | | | X | We shifted item 2 (Your thoughts and feelings) up to become the first item. |
| | Items 4 and 6 are related. | | | X | | | | We put items 4 (Your medication) and 6 (Your care) one after the other. |
| | Items 1 and 3 are related. | | | X | | | | We put items 1 (Your habits and activities) and 3 (Your situation) one after the other. |
| **Content of item 2** (Your thoughts and feelings) | All other answers depend on the acceptance or not of one's HIV status [which is addressed in item 2, Your thoughts and feelings]. | | | | | | X | In item 2, we made "acceptance of having HIV" the first example barrier provided. |
| **Content of item 3** (Your situation) | The item is hard to answer/too long, given the many issues covered within this domain. Similar issues (relationships and stigma vs housing and finances) can be grouped together in separate items. | X | | X | | | X | We divided item 3 into two items, one for social and stigma-related barriers and one for financial and housing-related barriers. |
| | The domain title "Your situation" is imprecise. Relationship and stigma barriers can be labelled "Your social situation." | | | X | | | | We qualified the domain title. As item 3 was divided into two items, their domain titles became "Your social situation" and Your economic situation." |
| **Content of item 5** (Your health) | The French expression 'santé globale' is preferable. | | | X | | | | In the French version, we replaced 'santé générale' by 'santé globale.' For consistency, in the English version, we changed 'general health' to 'overall health.' |
| | The item is difficult to answer or understand (or could be for some people). The meaning or role of the specific barriers, especially, that of 'infection-related symptoms' is not clear. The item should be rewritten. | X | X | | | X | | We reformulated the item to simplify the language and refer directly to 'HIV symptoms.' |

*Session 1: Sept. 23 2020 (4 anglophone participants in Quebec; 3 male and 1 female); Session 2: Sept. 24 2020 (2 francophone males in Quebec); Session 3: Oct. 5 2020 (1 francophone female in France); Session 4: Nov. 5 2020 (1 francophone male in Quebec); Session 5: Nov. 9 2020 (1 francophone female in Quebec); Session 6: Nov. 16 2020 (3 francophone males in France).

ART [39], measured by the average of three questions, using an ordinal scale from 1 (=strongly disagree) to 5 (=strongly agree), where Cronbach alpha for this instrument in our sample was 0.87; and 4) self-reported last viral load test result (a measure created for this study), using a nominal scale with three categories ("undetectable", "detectable", "I don't know"). For participants recruited at the Chronic Viral Illness Service of the McGill University Health Centre in Montreal, we extracted anonymized plasma HIV viral load test results (number of HIV copies per milliliter of blood) from the patients' medical files that were closest to each timepoint (retrieved March 20, 2024). Note that the median time elapsed between

the Time 1 and Time 2 plasma viral load test results for these participants was 123.5 days (approximately 4 months' duration). The choice of the dependent variables reflects the lack of a gold standard for ART adherence measurement [40] and for most PROMs [41], hence our inclusion of a mix of indirect 'subjective' and 'objective' [42] measures of adherence. The following categorical sociodemographic characteristics of the participants were also collected: selected survey language (English or French) as well as self-reported country of residence (Canada or France), immigration status (i.e., whether born outside the country of residence or not), age in years, level of education, sex given at birth, and sexual orientation. Furthermore, the mode of survey completion was recorded, especially, whether self- or other-administered.

**Main study variables.** In the logistic regression analyses (described in the next section), the I-Score's seven items served as independent variables and all five dependent variables were dichotomized: 1) ART adherence in the last 30 days was coded 1 if its value was greater or equal to 95% (a commonly used optimal ART adherence threshold) [43] and to 0 otherwise; 2) ART adherence in the last 7 days was coded 1 if its value was 1 (=all HIV pills taken) and 0 otherwise; 3) intention to adhere to ART was coded 1 if its value was an average of 5 and 0 if less than 5; 4) self-reported viral load was set to 1 if it was "undetectable" and 0 otherwise; and 5) plasma viral load test results were coded 1 if the number of HIV copies per mL of blood was less than 20 (recognized as an optimal level of viral suppression) [44] and 0 otherwise.

**Statistical analyses.** All analyses were performed with R (version 4.3.1) [45]. Descriptive statistics are provided for the participants' sociodemographic characteristics, mode of survey administration, and the independent and dependent variables, including when stratified by the characteristics of interest. Missing data are reported for all variables. All provided $p$-values respect standards for reporting and interpreting statistical significance in medical research [46].

As an indicator of structural validity, the redundancy of the I-Score's seven items was evaluated with inter-item Pearson correlations at Time 1, with strongly correlated items (i.e., a correlation coefficient of at least 0.80), considered for possible elimination [47]. In addition, investigating whether each I-Score item represents a unique and independent dimension of barriers to ART adherence, we performed a principal component analysis (PCA) on the inter-item correlation matrix at both Time 1 and Time 2. We considered seven principal components and extracted the corresponding loadings. If the associations between the principal components and the I-Score's seven items were difficult to interpret, we applied a varimax rotation. Structural validity was indicated if each of the I-Score's seven item was strongly (loading of ≥ 0.80) and independently associated with only one of the seven principal components.

To evaluate the concurrent validity of the I-Score's seven items, five logistic regressions were fitted to the data, one for each dichotomized dependent variable, using only Time 1 data. All seven I-Score items were considered covariates. Covariate significance was set at a $p$-value of $\alpha < 5\%$. We also calculated the receiver operating characteristic (ROC) curve for each logistic regression, and the corresponding area under the curve (AUC). To classify the predictive capacity of each logistic regression based on the corresponding AUC, we used the thresholds proposed by Hosmer et al. [48] ($0.5 < AUC < 0.70 = $ 'Poor'; $0.70 \leq AUC < 0.80 = $ 'Acceptable'; $0.8 \leq AUC < 0.9 = $ 'Excellent'; $AUC \geq 0.9 = $ 'Outstanding') and Swets [49] ($AUC: < 0.70 = $ 'Low'; $0.70 \leq AUC < 0.90$ 'Useful'; $AUC \geq 0.90 = $ 'High' accuracy). Evaluation of the I-Score's predictive validity proceeded as described for testing its concurrent validity, while using Time 2 data for the dependent variables. Subgroup analyses were performed to evaluate the sensitivity of the models to the participants' sociodemographic characteristics and the mode of survey administration (to determine measurement invariance).

To assess the I-Score's reliability, we employed a test-retest reliability approach. With the unrotated PCA performed on the inter-item correlation matrix at both Times 1 and 2 for the I-Score's structural validity, we summarized the information contained in its seven items by retaining principal components based on Kaiser's criterion (i.e., those corresponding to an eigenvalue greater than 1). We also validated the number of retained principal components with a scree plot, selecting the number that appears prior to the plot's elbow. We expected to find a single principal component that explains a large proportion of the variance in the data, with similar loadings across the I-Score's seven items. Once each participant's score was obtained by summing the products of each loading with the corresponding I-Score item's standardized response, test-retest reliability was then assessed with the intraclass correlation coefficient (ICC), computed from the obtained

scores. Following COSMIN recommendations [32], the ICC was calculated with a two-way random effects model, with good reliability indicated by an ICC ≥ 0.70.

To estimate measurement error, we followed COSMIN [32] recommendations to use the standard error of measurement (SEM) for continuous scores based on a test-retest design. The SEM, which is obtained from the ICC, has a possible range of 0–1, with a SEM closer to 0 considered more favorable.

**Working hypotheses and performance thresholds.** As mentioned, for our purposes, adequate structural validity would be indicated by Pearson inter-item correlation coefficients below 0.80 (absence of redundant items) as well as high (≥ 0.80) and independent factor loadings of the items on each of the 7 components of the PCA.

For issues of construct validity, our hypotheses were as follows:

a) With the global sample, at least one I-Score item (covariate) would be statistically associated with each dependent variable in at least 75% of the logistic regression analyses (excluding inconclusive results).

b) With the global sample, the I-Score's accuracy would be at least "Acceptable; Useful" (AUC ≥ 0.70) for at least 75% of the analyses for each dependent variable (excluding inconclusive results).

c) For each subgroup examined, the I-Score's accuracy would be at least "Acceptable; Useful" (AUC ≥ 0.70) for at least 75% of the analyses across the dependent variables (excluding inconclusive results).

The choice of a 75% threshold was based on COSMIN criteria for the sufficiency of results in hypothesis or responsiveness testing to determine a PROM's measurement properties [32].

Finally, to review, reliability would be indicated by an ICC of ≥ 0.70 and low measurement error, by a SEM closer to 0.

## Results

Table 2 shows the sample's sociodemographic information. A total of n = 305 participants were included in the analyses, 31% of whom were female, indicating that we surpassed our sample size and sex-based recruitment goals.

S1 and S2 Tables display the descriptive statistics of the previously described independent and dependent variables at Times 1 and 2 and stratified by the sample's sociodemographic characteristics, and the survey's mode of administration.

**Structural validity.** Table 3 shows the Pearson correlation coefficients for all pairs of the I-Score items. Coefficients ranged from 0.29 to 0.62. As none was greater than or equal to 0.80, no item was considered for elimination.

S3 and S4 Tables show the loadings of the seven principal components at Time 1 and Time 2 respectively, obtained after a varimax rotation. For both timepoints, each principal component was found to have only one high loading (from 0.84 to 0.95), with the other loadings being very low (from 0.10 to 0.28). This indicates that each of the I-Score's items is strongly and uniquely associated with only one principal component.

**Concurrent and predictive validity.** Table 4 presents the AUC's of the ten logistic regressions for the global sample and stratified by the considered sociodemographic characteristics and mode of survey administration to test the I-Score's concurrent (Time 1 to Time 1) and predictive (Time 1 to Time 2) validity. Inconclusive results are identified by an asterisk in Tables 4 and S5, which presents the odds ratios for the covariates of the logistic regressions with 95% confidence intervals. Inconclusive results were caused by a lack of convergence of the iterative procedure to obtain the regression coefficient estimates or by abnormally high values for the estimated standard errors of the regression coefficients, possibly due, in turn, to small subgroup sizes.

For hypothesis (a), the results of the ten logistic regression analyses with the full sample show that in 8 (80%), at least one I-Score item (covariate) emerged as statistically significant (S5 Table). Thus, we successfully attained our threshold. The three significant items were Habits and activities (5 analyses), Health [3], and Economic situation [1].

For hypothesis (b), again with the full sample, the results of the ROC analyses and AUC's indicate that at least "Acceptable; Useful" I-Score model performance (AUC ≥ 0.70) was obtained in at least 75% of analyses for self-reported viral load

**Table 2.  Sociodemographic characteristics of the sample (n = 305).**

|  | n (%) |
|---|---|
| **Survey language** |  |
| English | 83 (27.2%) |
| French | 222 (72.8%) |
| **Country of residence** |  |
| Canada | 261 (85.6%) |
| France | 41 (13.4%) |
| Missing | 3 (1.0%) |
| **Immigration status** |  |
| Immigrant | 161 (46.5%) |
| Non-immigrant | 142 (52.8%) |
| Missing | 2 (0.7%) |
| **Age (years)[1]** |  |
| < 50 | 135 (44.2%) |
| ≥ 50 | 160 (52.5%) |
| Missing | 10 (3.3%) |
| **Level of education** |  |
| Primary/Elementary | 19 (6.2%) |
| Secondary (High school)/Professional degree | 109 (35.8%) |
| College (post-secondary)/CEGEP/ Technical degree | 59 (19.3%) |
| University | 104 (34.1%) |
| Other | 8 (2.6%) |
| Missing | 6 (2.0%) |
| **Sex** |  |
| Female | 94 (30.8%) |
| Male | 207 (67.9%) |
| Missing | 4 (1.3%) |
| **Sexual orientation** |  |
| Heterosexual | 119 (39.0%) |
| Homosexual | 137 (44.9%) |
| Bisexual | 22 (7.2%) |
| Other | 5 (1.7%) |
| Prefer not to answer | 18 (5.9%) |
| Missing | 4 (1.3%) |

[1] Mean (SD) = 50.4 (13.0); Range: 19.0–82.0; for n = 295 participants.

(100%; AUC: 0.75–0.80) and plasma viral load test results (100%: AUC: 0.74–0.77). Results were below the threshold for ART adherence in the last 30 days (50%; AUC: 0.68–0.73), ART adherence in the last 7 days (50%; AUC: 0.66–0.72) and intention to adhere to ART (0%; AUC: 0.65–0.66). Hence, we achieved partial success for hypothesis (b).

For hypothesis (c), subgroup analyses generated the following percentage of AUC ≥ 0.70 results, not including inconclusive results: English survey respondents (100%; 8/8), French survey respondents (50%; 5/10), residents of Canada (70%; 7/10); residents of France (100%; 3/3), immigrants (70%; 7/10), non-immigrants (89%; 8/9), individuals aged less than 50 (86%; 6/7), individuals aged 50 or over (80%; 8/10), those with a high school/secondary education (78%; 7/9), the college-educated (83%; 5/6), the university-educated (100%; 8/8), females (57%; 4/7), males (90%; 9/10), heterosexuals (88%; 7/8), homosexuals (90%; 9/10), those completing the survey by face-to-face interview (100%; 7/7), and those

**Table 3. Pearson inter-item correlation matrix.**

| I-Score item | Thoughts and feelings | Habits and activities | Social situation | Economic situation | Medication | Care |
|---|---|---|---|---|---|---|
| Habits and activities | 0.51 | – | – | – | – | – |
| Social situation | 0.62 | 0.39 | – | – | – | – |
| Economic situation | 0.49 | 0.36 | 0.56 | – | – | – |
| Medication | 0.50 | 0.33 | 0.44 | 0.47 | – | – |
| Care | 0.44 | 0.29 | 0.43 | 0.39 | 0.36 | – |
| Health | 0.50 | 0.31 | 0.40 | 0.34 | 0.50 | 0.48 |

self-administering the survey online (100%; 10/10). Overall, in the subgroup analyses, for respondents completing the survey in French and females, the results were clearly below threshold. Results were considered borderline for Canadian residents and immigrants. Our requirements were, therefore, mostly met for hypothesis (c).

Considering the results obtained per dependent variable, across the global sample and subgroup analyses, large differences in the percentage of analyses indicating an acceptable performance of the I-Score models (i.e., AUC ≥ 0.70) were observed. For intention to adhere to ART, only 51% of AUC's were above threshold (18/35), compared with 83% for adherence in the past 30 days (30/36), 85% for adherence in the last 7 days (29/34), and 100% for both self-reported viral load (24/24) and plasma viral load test results (23/23). Without the intention to adhere variable, the 75% threshold for hypothesis (c) would be met for women (80%; 4/5), but not for French survey respondents (63%; 5/8). The results for Canadian residents (88%; 7/8) and immigrants (75%; 6/8) would also surpass the target and no longer be borderline.

Additionally, across the subgroup analyses, the percentage of AUC's ≥ 0.70 (not considering inconclusive results) was 84% (62/74) for concurrent validity testing (baseline data only) and 82% (56/68) for analyses assessing predictive validity. Equal percentages were found for analyses with the global sample (i.e., 60% in both cases), further suggesting similar accuracy with concurrent and future outcomes, as evaluated.

**Reliability.**  Table 5 shows the eigenvalues extracted from the correlation matrix, for each principal component, at each measurement time. The eigenvalues greater than 1, according to Kaiser's criterion, are highlighted in bold.

As expected, we retained only the first principal component at each timepoint. The scree plots at Time 1 and Time 2 supported this decision. This component explained 52% and 61% of the variance in the data at Time 1 and Time 2, respectively. Table 6 shows the loadings on the retained principal component of the I-Score's seven items, for each timepoint.

The loadings are similar for each timepoint, indicating a roughly balanced contribution of each item to score computation. Finally, we obtained an ICC of 0.81, which is above the threshold defined by COSMIN standards, establishing the measure's reliability.

**Measurement error.**  As to measurement error, we obtained a SEM of 0.43 which is closer to 0 than 1, as preferred.

## Discussion

Given its perceived utility, our team set out to create a short, yet comprehensive, patient-reported measure of barriers to ART adherence for administration as a part of routine HIV care (the 7-item I-Score). This manuscript summarizes the evidence accumulated to date for its use in urban cities in Canada (mainly Montreal and Toronto) and France (Paris) among adults with HIV on ART. We briefly described the instrument's development, including conceptual framework generation, but mainly presented new results obtained from cognitive testing and measurement property assessment. In the process, we showcased evidence of the measure's content validity, construct validity, and reliability, as well as on its measurement error.

### Conceptual framework

The previously published conceptual framework [24], grounded in the results of 41 qualitative studies among people living with HIV in developed countries, provided a strong foundation from which to generate measure items. Encompassing

**Table 4. Areas Under the ROC Curve (AUC) of the ten logistic regressions, with the interpretation thresholds of Hosmer et al. (H) and Swets (S), for the global sample and stratified by sociodemographic group and mode of survey administration.**

| Sample or group | Dependent variable | | | | | | | | | | |
|---|---|---|---|---|---|---|---|---|---|---|---|
| | Data | ART adherence: last 30 days | | ART adherence: last 7 days | | Intention to adhere to ART | | Viral load: self-reported | | Viral load: plasma | |
| | | AUC | n (missing) | AUC | n (missing) | AUC | n (missing) | AUC | n (missing) | AUC | n (missing) |
| **Global sample** (n=305) | Time 1 | 0.68 | 284(21) | 0.72 | 297(8) | 0.66 | 296(9) | 0.80 | 285(20) | 0.74 | 108(197) |
| | Time 2 | 0.73 | 230(75) | 0.66 | 241(64) | 0.65 | 241(64) | 0.75 | 234(71) | 0.77 | 72(233) |
| **Survey language** | | | | | | | | | | | |
| English (n=83) | Time 1 | 0.86 | 79(4) | 0.83 | 81(2) | 0.74 | 81(2) | 0.88 | 76(7) | 0.77 | 36(47) |
| | Time 2 | 0.90 | 59(24) | 0.90 | 62(21) | 0.73 | 62(21) | * | 59(24) | * | 27(56) |
| French (n=222) | Time 1 | 0.66 | 205(17) | 0.67 | 216(6) | 0.66 | 215(7) | 0.78 | 209(13) | 0.72 | 72(150) |
| | Time 2 | 0.66 | 171(51) | 0.70 | 179(43) | 0.65 | 179(43) | 0.80 | 175(47) | 0.77 | 45(177) |
| **Country of residence** | | | | | | | | | | | |
| Canada (n=261) | Time 1 | 0.70 | 241(20) | 0.73 | 253(8) | 0.65 | 252(9) | 0.80 | 241(20) | 0.74 | 108(153) |
| | Time 2 | 0.75 | 195(66) | 0.67 | 205(56) | 0.61 | 205(56) | 0.76 | 197(64) | 0.77 | 72(198) |
| France (n=41) | Time 1 | 0.73 | 41(0) | * | 41(0) | 0.85 | 41(0) | * | 41(0) | – | 0(0) |
| | Time 2 | 0.91 | 33(8) | * | 34(7) | * | 34(7) | * | 34(7) | – | 0(0) |
| **Immigration status** | | | | | | | | | | | |
| Immigrant (n=161) | Time 1 | 0.68 | 146(15) | 0.81 | 156(5) | 0.65 | 154(7) | 0.86 | 147(14) | 0.75 | 70(91) |
| | Time 2 | 0.70 | 109(52) | 0.68 | 117(44) | 0.70 | 117(44) | 0.72 | 114(47) | 0.85 | 41(120) |
| Non-immigrant (n=142) | Time 1 | 0.74 | 136(6) | 0.82 | 139(3) | 0.70 | 140(2) | 0.78 | 136(6) | * | 38(104) |
| | Time 2 | 0.72 | 120(22) | 0.74 | 123(19) | 0.68 | 123(19) | 0.88 | 118(24) | 0.82 | 31(111) |
| **Age (years)** | | | | | | | | | | | |
| < 50 (n=135) | Time 1 | 0.66 | 124(11) | 0.80 | 131(4) | 0.72 | 131(4) | * | 126(9) | 0.84 | 47(88) |
| | Time 2 | 0.82 | 104(31) | 0.71 | 106(29) | 0.71 | 105(30) | * | 104(31) | * | 34(101) |
| ≥ 50 (n=160) | Time 1 | 0.74 | 152(8) | 0.80 | 156(4) | 0.65 | 155(5) | 0.90 | 150(10) | 0.72 | 57(103) |
| | Time 2 | 0.70 | 118(42) | 0.70 | 127(33) | 0.61 | 128(32) | 0.83 | 121(39) | 0.80 | 37(123) |
| **Education level** | | | | | | | | | | | |
| Secondary (high school) or professional degree (n=109) | Time 1 | 0.70 | 104(5) | 0.70 | 107(2) | 0.62 | 107(2) | 0.86 | 102(7) | 0.75 | 37(72) |
| | Time 2 | 0.90 | 75(34) | 0.78 | 83(26) | 0.65 | 83(26) | 0.80 | 81(28) | * | 25(84) |
| College (post-secondary), CEGEP or technical degree (n=59) | Time 1 | 0.70 | 53(6) | 0.84 | 55(4) | 0.68 | 55(4) | * | 52(7) | * | 21(38) |
| | Time 2 | 0.77 | 41(18) | 0.85 | 44(15) | 0.70 | 44(15) | * | 41(18) | * | 12(47) |
| University (n=104) | Time 1 | 0.78 | 97(7) | 0.84 | 103(1) | 0.78 | 103(1) | * | 101(3) | 0.81 | 41(63) |
| | Time 2 | 0.81 | 89(15) | 0.84 | 89(15) | 0.71 | 89(15) | * | 86(18) | 0.77 | 31(73) |
| **Sex** | | | | | | | | | | | |
| Female (n=94) | Time 1 | 0.75 | 78(16) | 0.82 | 88(6) | 0.60 | 87(7) | 0.90 | 83(11) | * | 29(65) |
| | Time 2 | 0.77 | 60(34) | 0.64 | 64(30) | 0.67 | 65(29) | * | 62(32) | * | 14(80) |
| Male (n=207) | Time 1 | 0.67 | 202(5) | 0.74 | 205(2) | 0.70 | 205(2) | 0.80 | 198(9) | 0.78 | 77(130) |
| | Time 2 | 0.72 | 167(40) | 0.70 | 174(33) | 0.71 | 173(34) | 0.71 | 168(39) | 0.83 | 56(151) |
| **Sexual orientation** | | | | | | | | | | | |
| Heterosexual (n=119) | Time 1 | 0.77 | 109(10) | 0.81 | 113(6) | 0.70 | 112(7) | 0.83 | 112(7) | 0.78 | 34(85) |
| | Time 2 | 0.82 | 83(36) | 0.73 | 88(31) | 0.67 | 88(31) | * | 86(33) | * | 19(100) |
| Homosexual (n=137) | Time 1 | 0.72 | 132(5) | 0.73 | 135(2) | 0.72 | 135(2) | 0.73 | 133(4) | 0.73 | 51(86) |
| | Time 2 | 0.75 | 116(21) | 0.74 | 121(16) | 0.68 | 121(16) | 0.71 | 116(21) | 0.80 | 41(96) |

*(Continued)*

**Table 4.** (Continued)

| Sample or group | Dependent variable | | | | | | | | | | |
|---|---|---|---|---|---|---|---|---|---|---|---|
| | Data | ART adherence: last 30 days | | ART adherence: last 7 days | | Intention to adhere to ART | | Viral load: self-reported | | Viral load: plasma | |
| | | AUC | n (missing) | AUC | n (missing) | AUC | n (missing) | AUC | n (missing) | AUC | n (missing) |
| **Survey administration** | | | | | | | | | | | |
| Face-to-face interview (n = 69) | Time 1 | 0.83 | 66(3) | 0.88 | 69(0) | 0.72 | 68(1) | * | 65(4) | * | 15(54) |
| | Time 2 | 0.90 | 47(6) | 0.73 | 47(6) | 0.82 | 47(6) | 0.77 | 47(6) | * | 2(51) |
| Self-administered online (n = 184) | Time 1 | 0.72 | 175(9) | 0.78 | 179(5) | 0.72 | 179(5) | 0.77 | 172(12) | 0.80 | 62(122) |
| | Time 2 | 0.75 | 152(14) | 0.72 | 163(3) | 0.72 | 163(3) | 0.85 | 157(9) | 0.91 | 65(101) |
| **Legend** | **H** | **S** | | | | | | | | | |
| | Poor | Low | | | | | | | | | |
| | Acceptable | Useful | | | | | | | | | |
| | Excellent | Useful | | | | | | | | | |
| | Outstanding | Highly accurate | | *Inconclusive results, H = Hosmer & Lemeshow, S = Swets. | | | | | | | |

**Table 5. Eigenvalues of the correlation matrix for each Principal Component (PC), at Times 1 and 2.**

| Time | PC1 | PC2 | PC3 | PC4 | PC5 | PC6 | PC7 |
|---|---|---|---|---|---|---|---|
| Baseline | **3.63** | 0.80 | 0.69 | 0.64 | 0.50 | 0.42 | 0.33 |
| Week 4 | **4.25** | 0.68 | 0.54 | 0.47 | 0.44 | 0.34 | 0.28 |

ART adherence barriers reported by people living with HIV within countries beyond those of Canada and France in the framework provides a basis to argue for the I-Score's potential application or adaptation to other contexts of use. This is further supported by the considerable overlap between the domains we identified and those of the WHO's generic model of factors affecting adherence to long-term therapy [26].

That the I-Score measures barrier domains is a unique feature of our instrument relative to other ART adherence barrier measures. While the basis of its comprehensiveness and conciseness, assessing domains does not provide information on the patient's precise barrier(s). For instance, with this measure, a patient will rate the frequency of adherence difficulty within the domain of Thoughts and feelings which includes barriers tied to "acceptance of having HIV, emotions (feeling sad, anxious, etc.), medication-related knowledge and beliefs or motivation to take medication." This is intentional. Our PROM offers a relatively quick screen meant to instigate patient-healthcare provider communication, a well-supported outcome of PROM use in clinical practice [50]. Discussion is expected, in turn, to pinpoint barriers and improve their management.

## Cognitive testing

We consider that the cognitive testing performed on the instrument conforms to COSMIN standards for such content validation studies [29]. Importantly, it identified several issues that could negatively impact the measure's ease of use, logical flow, and comprehensibility. As a result, we integrated participant-recommended gradations to the response scale, from 0 to 10, with a clear midpoint. Such visual analogue scales have been used for over a century and are common metrics for PROMs [51], including in ART adherence measurement [52]. The choice of this response scale is further supported by their reported advantages for health state measurement, including their simplicity, validity, reliability, practicality, feasibility,

**Table 6. Loadings between the first Principal Component and the I-Score's seven items, at Times 1 and 2.**

| Time | Thoughts and feelings | Habits and activities | Social situation | Economic situation | Medication | Care | Health |
|------|----------------------|----------------------|------------------|--------------------|------------|------|--------|
| Baseline | 0.82 | 0.62 | 0.78 | 0.72 | 0.71 | 0.67 | 0.70 |
| Week 4 | 0.81 | 0.73 | 0.77 | 0.78 | 0.80 | 0.79 | 0.77 |

and acceptability [53]. Another significant change made to the I-Score was the addition of an item (due to separating an existing item (i.e., Your situation) into two (Your social situation, Your economic situation)). In a recent Cochrane review of ART adherence measures [9] intended to help identify instruments that could be used in resource limited clinical settings to identify patients in need of adherence support, instruments with more than 8 items were excluded. This suggests that, at 7 items, the I-Score's length remains feasible for use in HIV care. As to the measure's recall period, it was set at the past 4 weeks, a timeframe recommended for self-reported ART adherence measures [54]. However, several participants found this period too short for barriers to arise. It was nevertheless maintained to concord with the recall period (past 30 days) of one of the adherence measures used to evaluate the I-Score's measurement properties. In the future, when seeking the measure's formal implementation in HIV care, it will be reconsidered by the team in collaboration with stakeholders. Finally, no comprehensiveness issues were raised in relation to the measure. While in two separate data collection sessions, participants mentioned the absence of sexuality, and in one, the absence of religion, further discussion revealed that they were not understood as barriers to ART adherence but as important aspects of life that they did not see represented in the instrument.

## Measurement properties

The statistical analyses conducted support the retention of each item of the I-Score. The inter-item correlation coefficients were generally in the weak to moderate range [55] and a PCA on the 7 items with a varimax rotation produced a set of 7 principal components, where each item was highly and independently associated with each component (a one-to-one correspondence). These findings argue for the dimensional independence of the items, thus providing evidence of the measure's structural validity.

Results on the concurrent and predictive validity of the I-Score highlight its limited accuracy in classifying people living with HIV based on their intention to adhere to ART. Results on the relationship between intention and medication adherence have been mixed [56] and, therefore, this finding may not be surprising. Arguably, behavioral adherence and viral load outcomes are of greater interest within the context of HIV care, and considering this, our success thresholds were largely met. Notably, with the global sample as well as in the subgroup analyses, 100% of conclusive results were above threshold (i.e., AUC ≥ 0.70) for self-reported HIV viral load (collected from the full sample) and plasma viral load test results (collected from a subsample of Montreal participants). These findings are encouraging, given the high variation observed in the ability of self-report measures of adherence to predict viral non-suppression in people with HIV [9]. The subgroup analyses also provide evidence of the cross-cultural validity of our instrument, which is rarely evaluated in adherence-related PROMs, despite the generation of translations [57].

With an ICC of 0.81, our measure surpassed COSMIN's minimum threshold for adequate reliability and we reported its measurement error (SEM = 0.43), significant because the measurement error of patient-reported outcome measures of medication adherence is rarely provided [57].

## Limitations

The 7-item I-Score's measurement properties were assessed with a limited, albeit diverse, sample of people with HIV, largely from Montreal, Toronto, and Paris. Efforts are currently underway to obtain funding for a Canada-wide study with a more substantial sample of people living with HIV and pharmacy refill data for further validation research. Notably, this will

help us compensate for many of the inconclusive results obtained in the regression analyses due, in part, to small sub-group sizes.

No global scoring scheme has yet been determined for the measure, and it is currently being piloted in HIV care in Montreal as a string of 7 independent items [58]. Likewise, a meaningful cutoff for the response scales to indicate a clinically significant barrier is also not available. These issues limit our instrument's interpretability and will be the focus of future research.

Furthermore, the subgroup analyses suggest that the performance of the French version of the I-Score within Canada is suboptimal for predicting concurrent or future self-reported adherence (but not self-reported or plasma HIV viral load). To better understand this finding, further cognitive testing of the instrument among francophones in Canada may be required. Further such testing may also be advisable to verify the I-Score's appropriateness among additional key populations (e.g., individuals who inject drugs, Indigenous peoples [1]) and account for individuals with low treatment satisfaction (all cognitive testing participants reported being satisfied with their ART regimen).

As this measure was developed for use in English or French in high-resource countries, its applicability to low-income countries where the burden of HIV is highest [59], requires investigation and cross-cultural validation.

Finally, the development of this measure began before the emergence of long-acting (LA) ART for the treatment of HIV. LA-ART may resolve a number of adherence barriers for some people living with HIV (e.g., daily oral pills as a reminder of living with HIV or as carrying the risk of unintended disclosure of HIV status), while potentially giving rise to new ones (e.g., anxiety around injections) [60]. Cognitive and measurement property testing of the instrument with individuals on LA regimens may be needed to ensure fit with this context of use.

## Conclusion

The 7-item I-Score, a patient-reported measure of ART adherence barriers for use in HIV care, has good evidence of content validity, construct validity, and reliability, as well as acceptable measurement error. Its associations with self-reported and plasma viral load, overall, and across a range of sociodemographic groups, appear especially strong. Further research is needed to determine clinically meaningful scoring and cutoff thresholds to facilitate interpretability. Nevertheless, in a context where annual PROM administration is being recommended by HIV clinical guidelines to all people living with HIV [61] and consensus is lacking on which measures to employ [62,63], the I-Score offers a solid option for use in adherence management. Contrary to many other barrier measures [18], it is designed for HIV care, it is comprehensive with few items, and it does not require the admission or presence of missed doses to score highly. Thus, it may lead to less stigmatizing discussions and more preventative interventions. We expect the results of our clinical pilot study with the I-Score [58] to be informative in these regards.

### Strengths and limitations of this study

- This project employed a multi-phase design to generate a range of qualitative and quantitative evidence in support of a new patient-reported outcome measure.

- It produced evidence of cross-cultural validity and the standard error of measurement, which are uncommonly reported measurement properties of adherence-related patient-report instruments.

- The small survey sample size of certain socioeconomic subgroups contributed to some inconclusive results in the construct validity analyses (areas under the curve).

- Establishing the generalizability of the results to all people living with HIV in each country (Canada and France) requires further investigation.

## Supporting information

**S1 File. Cognitive interview schedule for the focus groups.**
(DOCX)

**S1 Table. Descriptive statistics of the I-Score's 7 items at each time point, for the global sample and stratified by sociodemographic group and mode of survey administration.**

(DOCX)

**S2 Table. Descriptive statistics of the five dependent variables at each time point, for the global sample and stratified by sociodemographic group and mode of survey administration.**

(DOCX)

**S3 Table. Loadings at baseline (Time 1) between the seven principal components obtained by a varimax rotation and the I-Score's seven items.**

(DOCX)

**S4 Table. Loadings at week 4 (Time 2) between the seven principal components obtained by a varimax rotation and the I-Score's seven items.**

(DOCX)

**S5 Table. Odds ratios for the seven covariates (with 95% confidence intervals) of the ten logistic regressions, for the global sample and stratified by sociodemographic group and mode of survey administration.**
(DOCX)

## Acknowledgments

The authors warmly thank Dr. Golriz Pahlavan and Alexandre Aslan, for their unwavering belief in and support of this project since its inception; Hayette Rougier for her invaluable help with recruitment and administration of the Paris study site; Francesco Avallone for assistance with qualitative coding of the cognitive interviews; and all the people living with HIV, healthcare professionals, and support staff who contributed to this work.

## Author contributions

**Conceptualization:** Kim Engler, Bertrand Lebouché.

**Data curation:** Serge Vicente.

**Formal analysis:** Kim Engler, Serge Vicente.

**Funding acquisition:** Kim Engler, Karine Lacombe, Bertrand Lebouché.

**Investigation:** Kim Engler, David Lessard.

**Methodology:** Kim Engler, Serge Vicente.

**Project administration:** David Lessard, Darrell HS Tan, Karine Lacombe.

**Supervision:** Kim Engler, Darrell HS Tan, Bertrand Lebouché.

**Writing – original draft:** Kim Engler, Serge Vicente.

**Writing – review & editing:** Kim Engler, David Lessard, Serge Vicente, Darrell HS Tan, Karine Lacombe, Bertrand Lebouché.

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
