## [Decision Letter · Decision Letter 0]

8 Dec 2024

Dear Dr. Engler,

Thank you for submitting your manuscript to PLOS ONE. After careful consideration, we feel that it has merit but does not fully meet PLOS ONE’s publication criteria as it currently stands. Therefore, we invite you to submit a revised version of the manuscript that addresses the points raised during the review process.

We look forward to receiving your revised manuscript.

Kind regards,

Kwasi Torpey, MD PhD MPH

Academic Editor

PLOS ONE

Journal Requirements:

2. Thank you for stating the following financial disclosure: [This work received support from the Canadian Institutes of Health Research (CIHR) HIV Clinical Trials Network (CTN) (CTN283). It was conducted within the program of a SPOR Mentorship Chair in Innovative Clinical Trials awarded by the CIHR to BL (Grant No. 383427). It benefited from methodological expertise and funding from the Quebec SPOR Support Unit -McGill Methodological Developments Platform (Grant number M006). It was also supported, in part, by a research grant from the Investigator Initiated Studies Program of Merck Canada Inc. (Grant number IISP-53538), which had no hand in the design, conduct, or writing up of this work. French (France) participation was supported, in part, by funding from MSD Avenir awarded to KL and managed by the AP-HP Foundation (MSDAVENIR DS-2018-0072).]. Please state what role the funders took in the study. If the funders had no role, please state: "The funders had no role in study design, data collection and analysis, decision to publish, or preparation of the manuscript." If this statement is not correct you must amend it as needed. Please include this amended Role of Funder statement in your cover letter; we will change the online submission form on your behalf.

3. Please expand the acronym “SPOR” (as indicated in your financial disclosure) so that it states the name of your funders in full. This information should be included in your cover letter; we will change the online submission form on your behalf.

4. Thank you for stating the following in the Competing Interests section: [BL has received research support and consulting fees from ViiV Healthcare, Merck, and Gilead. KL has received travel support and consulting fees from ViiV Healthcare, Merck, and Gilead. DHST’s institution has received research support for investigator-initiated research studies from Abbvie and Gilead, and for participation in industry-sponsored trials from Glaxo Smith Kline. The other authors have no competing interests to declare.]. Please confirm that this does not alter your adherence to all PLOS ONE policies on sharing data and materials, by including the following statement: "This does not alter our adherence to PLOS ONE policies on sharing data and materials.” (as detailed online in our guide for authors http://journals.plos.org/plosone/s/competing-interests). If there are restrictions on sharing of data and/or materials, please state these. Please note that we cannot proceed with consideration of your article until this information has been declared. Please include your updated Competing Interests statement in your cover letter; we will change the online submission form on your behalf.

5. We noted in your submission details that a portion of your manuscript may have been presented or published elsewhere. [In the background, we briefly summarize our past published work on the development of the I-Score measure (i.e., our needs assessment survey, our review of existing measures on barriers to ART adherence, and the development of our conceptual framework).] Please clarify whether this [conference proceeding or publication] was peer-reviewed and formally published. If this work was previously peer-reviewed and published, in the cover letter please provide the reason that this work does not constitute dual publication and should be included in the current manuscript.

6. In the online submission form, you indicated that your data will be submitted to a repository upon acceptance. We strongly recommend all authors deposit their data before acceptance, as the process can be lengthy and hold up publication timelines. Please note that, though access restrictions are acceptable now, your entire minimal dataset will need to be made freely accessible if your manuscript is accepted for publication. This policy applies to all data except where public deposition would breach compliance with the protocol approved by your research ethics board. If you are unable to adhere to our open data policy, please kindly revise your statement to explain your reasoning and we will seek the editor's input on an exemption.

7. Your ethics statement should only appear in the Methods section of your manuscript. If your ethics statement is written in any section besides the Methods, please move it to the Methods section and delete it from any other section. Please ensure that your ethics statement is included in your manuscript, as the ethics statement entered into the online submission form will not be published alongside your manuscript.

Reviewers' comments:

Reviewer's Responses to Questions

**Comments to the Author**

1. Is the manuscript technically sound, and do the data support the conclusions?

Reviewer #1: Yes

Reviewer #2: Yes

Reviewer #3: Partly

Reviewer #4: Partly

2. Has the statistical analysis been performed appropriately and rigorously?

Reviewer #1: Yes

Reviewer #2: Yes

Reviewer #3: I Don't Know

Reviewer #4: Yes

3. Have the authors made all data underlying the findings in their manuscript fully available?

Reviewer #1: Yes

Reviewer #2: Yes

Reviewer #3: No

Reviewer #4: Yes

4. Is the manuscript presented in an intelligible fashion and written in standard English?

Reviewer #1: Yes

Reviewer #2: Yes

Reviewer #3: Yes

Reviewer #4: Yes

Reviewer #1: Thank you for the opportunity to review this manuscript. The authors have done a great job in writing this manuscript; it is well written. There are a few areas that I'd seek clarity on and I list them below:

1.Line 84: The authors make an observation that despite tremendous progress with the UNAID 95-95-95 goals, attainment of viral suppression remains a challenge for individuals who inject drugs, and Indigenous peoples in Canada. However the two aforementioned groups wee not included among the main interest groups during cognitive testing (line 188 and 208) This presents a tension and I am curious how the 7-item score developed offers utility for ART adherence (despite its limited generalizability) to these two important sub-populations.

2. Line 131: Use of the term 'people with HIV' rather than persons living with HIV (PLHIV). Recommend that the authors update the language in the manuscript to be consistent with updated language in the review the UNAIDS terminology guidelines of 2024.

3. Line 25-26: The overlap between the authors' conceptual framework and WHO's model of factors affecting adherence to long-term therapy seems to be very minute. That the WHO's model has wide acceptability and used in many countries, it would have been important to understand why the authors' discount the generalizability of their tool (yet it is very close to what WHO has developed). Expounding on the same would be necessary.

4. Line 178: In assessing the barrier domains of medication-taking, it would be important for the authors to document how they account for advances in antiretroviral treatment simplification that has resulted in a single pill per day and in some cases for children improved taste of medication eg among childhood ART molecules.

Questions on the measurement properties section.

5. The 7-item I-Score assesses barrier domains with 42 additional items on specific barriers identified. These 42 additional items used for the PCA sample size calculations seem to be nested within the 7 domains. How did the authors handle this analysis?

6. Time for data collection – baseline and 4 weeks later. What programmatic/clinical reason informs the 4 weeks later? Is it sufficient for adherence assessment? Other than for test-retest method on lines 310-319, how was the longitudinal nature of the data taken care of? Did the author consider correlation of the responses from an individual between the 2 time points?

7. Intention to adhere question – 3 questions are used to make an average on an ordinal scale. Doesn't averaging lead to a lot of information being lost? Would the authors consider using one question that carries more weight on intention to adhere?

Reviewer #2: This is a well-written paper that addresses a critical gap in HIV care by developing a validated tool to assess and manage ART adherence barriers. Below are specific comments to further strengthen the manuscript:

Conceptual framework (Line 148 -181)

This section presents the conceptual framework for the I-Score development in detail but could benefit from addressing the following:

While the qualitative synthesis and thematic analysis are mentioned, the description of how the conceptual framework was developed could be clearer. For example, was the thematic analysis inductive, deductive, or a combination? What criteria determined the inclusion of the 41 studies?

Saturation levels (66%-100%) are mentioned, but it’s unclear how this was calculated or why this range is acceptable.

The overlap and distinctions between the developed framework and the WHO model are noted, but the implications of these differences are not well-explored. Please explain how these distinctions (e.g., separating cognitive and emotional aspects, reclassifying stigma) enhance the utility or validity of your framework.

The decision to include one item per domain for the measure is justified by feasibility concerns, but there’s limited discussion of how this impacts content validity. Reducing 70 distinct barriers to six items might overlook nuances within each domain. Please discuss how the single-item approach balances brevity with comprehensiveness. Also, please consider addressing any potential limitations in capturing the complexity of adherence barriers.

The framework’s applicability beyond developed countries is not discussed. Acknowledge this limitation and suggest how the framework could be adapted or tested in other settings such in under developed countries.

Results

• The geographic distribution is uneven (e.g., five from Montreal, one from Pointe-Claire, and four from Paris), but the implications of this imbalance are not addressed.

• The sample lacks gender balance (9 males, 3 females), and there is no mention of whether this aligns with the target demographic for the PROM.

• The demographic and geographic specifics are presented without reflection on generalizability. For example, did participants' varied education levels, languages, or cultural contexts affect the feedback received?

• Cognitive Testing Methodology: Were participants asked to explain their thought processes for each item? Were any techniques used to evaluate response scale comprehension or item relevance?

• Intention to adhere: The significantly lower AUC (51%) for "intention to adhere" is reported but not explained. This result could indicate a limitation of the I-Score or of intention as a construct. PLease describe possible reasons for this lower performance. For example, does intention to adhere poorly predict actual adherence? Is the I-Score less effective in capturing this construct? Also, please discuss how the I-Score could be improved to address weaker areas like intention to adhere or subgroup disparities.

Reviewer #3: This is an interesting body of work and i congratulate the authors for putting the manuscript together.

My most pressing concern with this manuscript is that it appears to be part of a piecemeal publication of the findings from a body of work. There were at least 7 obvious self citations of publications of outputs from largely the same process.

Secondly, the authors introduce elements to the results that were not included or described in the methods. There were also instances of approaches that may have introduced bias to the results e.g. convenience sampling, changes made or withheld for the reason of being pragmatic, and cut-offs that had no scientific explanation other than that they are commonly used. Although the authors describe in their limitation, the impact of the small sample size on the outputs from the multiple logistic regressions, it calls to question, the utility of the outputs in this regard.

The authors arguments around lack of consensus on adherence/barriers measures seem to be a reason to not have just one more added to the fray. This work does not convince me to be addressing that gap. In essence, it is only introducing yet another measure. It seems, that there was also no deliberate attempt to include as participants, the very groups that the authors identify as in most need of measures of barriers to adherence.

These and a few more specific comments have been included in the attached file.

Reviewer #4: Reviewers report

To: PLOS ONE

Manuscript: A multi-phase project to develop a patient-reported measure of barriers to antiretroviral therapy adherence for use in HIV care: The 7-Item I-Score (PONE-D-24-21594)

General comment:

This is great work and will have massive contribution to the HIV prevention and control in both countries and other high-income countries. However, it lacks data from low income countries where those countries have high contribution to the Global burden of death due to HIV. So, the rationale on How the two countries (Canada and France) were selected for this tool development is not clear.

Specific comments.

Abstract: Minimize acronym sin the abstract; line 33: objectives=> Objective: delete ‘s’ and Methods of the abstract lack richness on the data source. What was the source of the tool? From older version of the adherence tool? Literature reviewed? Just a qualitative research synthesis is not clear to readers. AND How were the 41 research articles searched and identified? Was its systematic approach?

Introduction: The rationale of this work is not clear, except just being originated from previous work which is great but this tool validation from experts in low resource setting like Africa is crucial for applicability. The brief distribution of the I-Score, conceptual framework has good insight to readers.

Methods: Very packed methodological details, so it needs unpacking the approaches that the authors used to yield such a great work. The mix of objective and methods is not clear. Good to link the objective with conceptual framework and then method independently, and Good to prepare a summary table that consist of Objectives with domains that may avoid redundant presentation of the objective.

Results: Good to move some of the information to ‘Methods section’, for example: line 200-210: the number of interviews, focus group discussion and other similar descriptions. Good to have a flow diagram that show the step-by-step development of the tool (Figure 1: should address the Review of literature/synthesis, Interviews and discussions, Framework developments, review by high level experts, piloting and quantitative data analysis for reliability and validity and the final validated tool).

The high- and low-income countries context is not even disclosed in the limitation section.

**Do you want your identity to be public for this peer review?** For information about this choice, including consent withdrawal, please see our Privacy Policy

Reviewer #1: No

Reviewer #2: No

Reviewer #3: No

Reviewer #4: No

---

## [Author Response · Author response to Decision Letter 1]

17 Mar 2025

All responses to the editor and to the reviewers are included in the attached files labelled Cover Letter and Response to Reviewers.

---

## [Decision Letter · Decision Letter 1]

8 Aug 2025

Dear Dr. Engler,

Thank you for submitting your manuscript to PLOS ONE. After careful consideration, we feel that it has merit but does not fully meet PLOS ONE’s publication criteria as it currently stands. Therefore, we invite you to submit a revised version of the manuscript that addresses the points raised during the review process.

We look forward to receiving your revised manuscript.

Kind regards,

Alejandro Torrado Pacheco, PhD

Staff Editor

PLOS ONE

Journal Requirements:

Additional Editor Comments:

The manuscript has been evaluated by three reviewers, and their comments are available below.

The reviewers make several recommendations to improve the clarity of the reporting, and have questions about some of the methodology.

Could you please revise the manuscript to carefully address the concerns raised?

Reviewers' comments:

Reviewer's Responses to Questions

**Comments to the Author**

Reviewer #1: All comments have been addressed

Reviewer #5: All comments have been addressed

Reviewer #6: (No Response)

2. Is the manuscript technically sound, and do the data support the conclusions?

Reviewer #1: Yes

Reviewer #5: Yes

Reviewer #6: Yes

3. Has the statistical analysis been performed appropriately and rigorously?

Reviewer #1: Yes

Reviewer #5: Yes

Reviewer #6: Yes

4. Have the authors made all data underlying the findings in their manuscript fully available?

Reviewer #1: Yes

Reviewer #5: Yes

Reviewer #6: Yes

5. Is the manuscript presented in an intelligible fashion and written in standard English?

Reviewer #1: Yes

Reviewer #5: Yes

Reviewer #6: Yes

Reviewer #1: The responses to reviewer comments are satisfactory. No further responses are required. I provide my concurrence for this work to be published.

Reviewer #5: Thank you for allowing me to review your very interesting and comprehensive manuscript. I appreciate the methodological rigour that went into the study and the development of the manuscript. It would be great to see the results when you scale it up after revising your I-score's french version as well as inclusion of other key populations.

I also appreciated your thoughtful approach to ensuring that the questions were non-stigmatizing or accusatory in any way to your clients.

Reviewer #6: General comments:

This manuscript includes the cognitive testing and evaluation of the measurement properties of the I-Score, a 7-item, self-reported, survey instrument used to assess antiretroviral therapy (ART) adherence barriers.

I found this manuscript to be somewhat dense, but overall a refreshing read. I say that because, at heart, the development of a predictive product. The standard these days is to use a massive dataset, throw some machine learning at it, and then have a deployable predictive model. Yet, I still feel there is value, and maybe tremendous value, in a concise instrument that can be administered efficiently to respondents and is able to capture most of the variability present in the data.

My biggest question is who the audience is for this manuscript? It is full of technical details, so it would seem that this is for the methodologists. That said, there are descriptions of some of the methods that suggest maybe this is for non-methods people. For instance, in lines 304-310 the description of PCA is probably not needed for a methodologist audience since they should be familiar with PCA and how PCA and rotations are used in survey development.

Furthermore, some of the details feel as though they could probably be moved to a supplement since they don't add too much to the overall story. For instance, Tables 4 and 5 are important, but do they add much? I felt like they could be referenced with a sentence or two in the text and included in the supplement for the supremely interested to see. Changes such as this would greatly help the flow of the manuscript and, I hope, not discourage the non-methodologists from reading this.

Specific comments:

1. (line 203) Is this supposed to be "...HIV in Canada"?

2. (lines 319-322) Is there any scientific basis for these thresholds or are they based on feels? These are pretty important to your assessments, but I don't find them to be that convincing. I'm not sure there is much you can change, but you might want to make note of this.

4. (Table 6) I found this table absolutely overwhelming. I'm not quite sure what to make of it and I'm not sure what to suggest. I guess my suggestion would be to create a forest-like plot with the AUCs for all the models (including 95% confidence intervals as error bars). You could color them based on the H and S interpretations if you wish. I can understand the desire to comment on the significant covariates, but I'm not the biggest fan of that. I encourage reporting the entire model, with all odds ratios and 95% CIs. That would have to be a supplementary table (or tables), but maybe you can include the significant variables in the text and then leave the details to the supplement.

5. (lines 678-680) If 75% was achieved in 100% of analyses, why is the lower AUC < 0.75? Similarly, why does a 50% have an AUC that is < 0.75? Sorry, I guess I am missing something with how these results are reported.

**Do you want your identity to be public for this peer review?** For information about this choice, including consent withdrawal, please see our Privacy Policy

Reviewer #1: No

Reviewer #5: **Yes: ** Gurpreet Kindra

Reviewer #6: No

---

## [Author Response · Author response to Decision Letter 2]

16 Sep 2025

Please refer to the response to reviewers in the attached file. It will provide a comprehensive description of the comments received and our answers.

---

## [Decision Letter · Decision Letter 2]

9 Dec 2025

A multi-phase project to develop a patient-reported measure of barriers to antiretroviral therapy adherence for use in HIV care: The 7-Item I-Score

PONE-D-24-21594R2

Dear Dr. Engler,

We’re pleased to inform you that your manuscript has been judged scientifically suitable for publication and will be formally accepted for publication once it meets all outstanding technical requirements.

Kind regards,

Marianne Clemence

Staff Editor

PLOS One

Additional Editor Comments (optional):

Reviewers' comments:

Reviewer's Responses to Questions

**Comments to the Author**

Reviewer #1: All comments have been addressed

Reviewer #6: All comments have been addressed

2. Is the manuscript technically sound, and do the data support the conclusions?

Reviewer #1: Yes

Reviewer #6: Yes

3. Has the statistical analysis been performed appropriately and rigorously?

Reviewer #1: Yes

Reviewer #6: Yes

4. Have the authors made all data underlying the findings in their manuscript fully available?

Reviewer #1: Yes

Reviewer #6: Yes

5. Is the manuscript presented in an intelligible fashion and written in standard English?

Reviewer #1: Yes

Reviewer #6: Yes

Reviewer #1: The responses by the authors are satisfactory. Kindly consider moving forward with publication of this manuscript.

Reviewer #6: (No Response)

**Do you want your identity to be public for this peer review?** For information about this choice, including consent withdrawal, please see our Privacy Policy

Reviewer #1: No

Reviewer #6: No

---

## [Editor Report · Acceptance letter]

PONE-D-24-21594R2

PLOS One

Dear Dr. Engler,

I'm pleased to inform you that your manuscript has been deemed suitable for publication in PLOS One. Congratulations! Your manuscript is now being handed over to our production team.

Kind regards,

on behalf of

Dr Marianne Clemence

Staff Editor

PLOS One